# Large Differences in Bud Burst and Senescence between Low- and High-Altitude European Beech Populations along an Altitudinal Transect in the South-Eastern Carpathians

Mihnea Ioan Cezar Ciocîrlan [1,2], Elena Ciocîrlan [1,*], Dănuț Chira [2], Gheorghe Raul Radu [2], Victor Dan Păcurar [1], Emanuel Beșliu [1,2], Ourania Grigoriadou Zormpa [3], Oliver Gailing [1,3] and Alexandru Lucian Curtu [1]

[1]   Faculty of Silviculture and Forest Engineering, "Transilvania" University of Brasov, 500123 Braşov, Romania; mihnea.ciocirlan@unitbv.ro (M.I.C.C.); vdpacurar@unitbv.ro (V.D.P.); besliu_emanuel@yahoo.com (E.B.); ogailin@gwdg.de (O.G.); lucian.curtu@unitbv.ro (A.L.C.)

[2]   National Institute for Research and Development in Forestry "Marin Drăcea", Braşov Research Station, 13 Cloşca Street, 500040 Braşov, Romania; danutchira61@gmail.com (D.C.); raulgradu@gmail.com (G.R.R.)

[3]   Department of Forest Genetics and Forest Tree Breeding, Büsgen-Institute Georg–August University of Göttingen, Büsgenweg 2, 37077 Göttingen, Germany; ourania.grigoriadouzormpa@uni-goettingen.de

*   Correspondence: ciocirlan.elena@unitbv.ro

**Abstract:** Phenology is considered an indicator of environmental changes, with direct implications in the length of the growing season; therefore, it offers essential information for a better understanding of the tree–environment relationships that could lead to the right decisions for forests' sustainable use and conservation. A better understanding of how European beech (*Fagus sylvatica*) phenology responds to predicted climate change effects is important for forest management. This study aimed to assess bud burst and senescence among and within beech populations located along a steep elevational gradient. Phenological observations were carried out on 150 beech individuals along an altitudinal transect in the south-eastern Carpathian Mountains, from 550 to 1450 m, in five study sites in two consecutive years. The start of the bud burst, of senescence, and the duration of the growing season varied inversely proportionally to the elevational gradient in both monitored years. Individuals located at the highest altitude need 28 more days to start the growing season than those at the lowest altitude. There is an average difference of 14 days at the start of the growing season in the same beech populations between the two consecutive years. The first stage of senescence (yellowing of leaves) lasted longer in 2021 (21–32 days) than in 2022 (18–25 days), with a difference of 16%–28%, proportional to the increase in altitude. The association of field phenological data with meteorological data indicates that the start of the growing season occurs when the thermal threshold of 10 °C is exceeded, with an accumulation of a least 60 GDD (growing degree days) with a threshold of 0 °C in the last 7 days as a complementary condition. The appearance of the first stage of senescence, the yellowing of the leaves, was also influenced by the temperature and the accumulation of at least 72 SDD (senescence degree days) with a threshold of 0 °C in the last 7 days. Our results confirm that the temperature is the triggering meteorological factor for the onset of bud burst and leaf senescence in European beech.

**Keywords:** leaf phenology; *Fagus sylvatica*; altitudinal transect; local adaptation

## 1. Introduction

Seasonal changes in terrestrial ecosystems are becoming more and more influenced by the effects of climate change, especially in the middle and higher latitudes [1]. In this context, the trend of temperature increase has become more noticeable. It can be directly linked with processes such as desertification, melting glaciers, reduced snow cover, intensifying heavy rainfall, and rising sea levels, and indirectly with soil erosion [2] and changing habitat areas for plants and animals. As a result of these effects of climate

change, species in forest ecosystems are forced to adapt and react through their regulatory mechanisms (physiological adaptation) or are even pushed to their survival limit [3].

Climate change alters the timing and length of the spring and autumn periods, which can significantly affect vegetation [4]. These seasonal changes in vegetation are determined by plant phenology [5], which is defined as the "synchronization of seasonal activities of plants and animals" [6,7]. The phenology of forest tree species is one of the most responsive and easily observable traits in response to climate effects. The survival rate, reproductive performance, persistence, and, therefore, the distribution range of a forest tree species are affected by phenological timing [8].

Temperature is a primary driver of forest tree species' growth and development. It influences the rates of chemical reactions in physiological processes [1], although its specific effects vary among organs [7]. Increases in air temperature due to the anthropogenic greenhouse effect can be detected easily in the phenological data of Europe within the last four decades [9]. In many cases, a higher temperature has been shown to accelerate a tree's development, with each degree increase in the spring temperature causing an advanced start of the leaf-out process by 2–7 days [5], which leads to an earlier transition to the next phenophase [1]. A longer growing season starts with advanced forest phenology driven by global warming [10]. Early spring and higher summer temperatures advance leaf yellowing [11]. On the contrary, warmer autumn temperatures delay leaf yellowing; an increase with each degree in autumn temperature causes a delayed senescence date by up to 8 days for some forest tree species [5]. The Earth's climate has warmed by approximately 0.6 °C over the past 100 years, with two main warming periods, between 1910 and 1945 and from 1976 onwards [6].

Taking into account only the crown condition (defoliation), European beech (*Fagus sylvatica* L.) is the healthiest major forest tree species in Europe [12], but it is one of the most sensitive hardwood species to uprooting produced by snow, ice, and wind [13,14]. European beech is Europe's most widespread forest tree species [15], including Romania (NFI Cycle II—[16]). It has high economic and ecological value, becoming a thoroughly investigated European tree species [17,18].

Aridity and warming have produced a decline in the growth rate of beech across Europe, except in the extreme north and high-altitude regions [19]. European beech decline was followed by severe droughts and heatwaves [20], flooding, and water excess [21,22]. Additional biotic factors (*Neonectria* sp. or *Phytophthora* sp.) have been aggravating factors of the decline [23,24]. Some ecological interactions between forest tree species and various communities of fungi and insects are influenced by the timing of leaf-out phenology [25].

Based on field observation and mathematical models, phenology can provide an algorithm/model to explain the reaction of forest tree species and the capacity to adapt to new site conditions. Even though field observations of tree phenology are labor-intensive, they offer valuable information regarding tree-level monitoring [26]. Time-series observations of spring phenology and senescence may lead to a better understanding of climate variability or climate change effects on plant responses as a direct correlation with meteorological data, especially temperature [27]. Understanding the impact of climate change on vegetation depends on how accurate the monitoring of phenology is and the precise delimitation of the start of the growing season and the end of it [28].

The onset of the bud burst of European beech is predominantly influenced by the seasonal course of temperature in late winter and early spring [29]. Prolonged droughts cause a premature onset of senescence in European beech, and this tendency is no longer only visible in low-altitude regions; it has also begun to appear in medium-altitude areas [30,31], directly involving a shorter growing season.

In the actual context of climate change, forest management must focus on species that have high adaptability and phenotypically plastic responses to new conditions, resistance to diseases and pests, and a wide range of uses for wood. Under climate change, the local adaptation and survival of European beech forests depend on their genetic variation and high adaptability to new environmental conditions [32]. These arguments justify the

choice of this species as a viable solution and the need to gain new information about its adaptation in as much detail as possible.

In this study, we aimed to assess the phenological variation in European beech along a complex altitudinal transect spanning 900 m of elevation during two years of monitoring through field observations and to associate these data with meteorological ones, especially temperature. The main objectives were to assess the phenological differences between the two consecutive years, among populations, between populations located at the extremities of the transect, and the intrapopulation variation, and to associate these data with the most appropriate meteorological indicator. The appearance of bud burst has already been linked with exceeding the threshold of 10 °C [33]; we evaluated the temperature variation (average daily temperature, and maximum and minimum daily temperature) from the last seven days until leaf flush. The association of temperature with the onset of senescence (yellowing of the leaves) was also evaluated.

## 2. Materials and Methods

### 2.1. Study Area

The study was conducted in the south-eastern Carpathian Mountains along an altitude transect at five study sites at an elevation between 550 and 1450 m (Figure 1; Table 1). One hundred fifty individuals were selected, thirty at each study site, located at a minimum distance of 25 m from each other and on north-facing slopes. This altitudinal transect with 900 m elevation overlapped with the natural range of *Fagus sylvatica* in the Brasov area, where it forms mixtures with other deciduous trees (*Acer pseudoplatanus* or *Carpinus betulus*) and conifers (*Picea abies* or *Abies alba*) with an age range of 80–120 years.

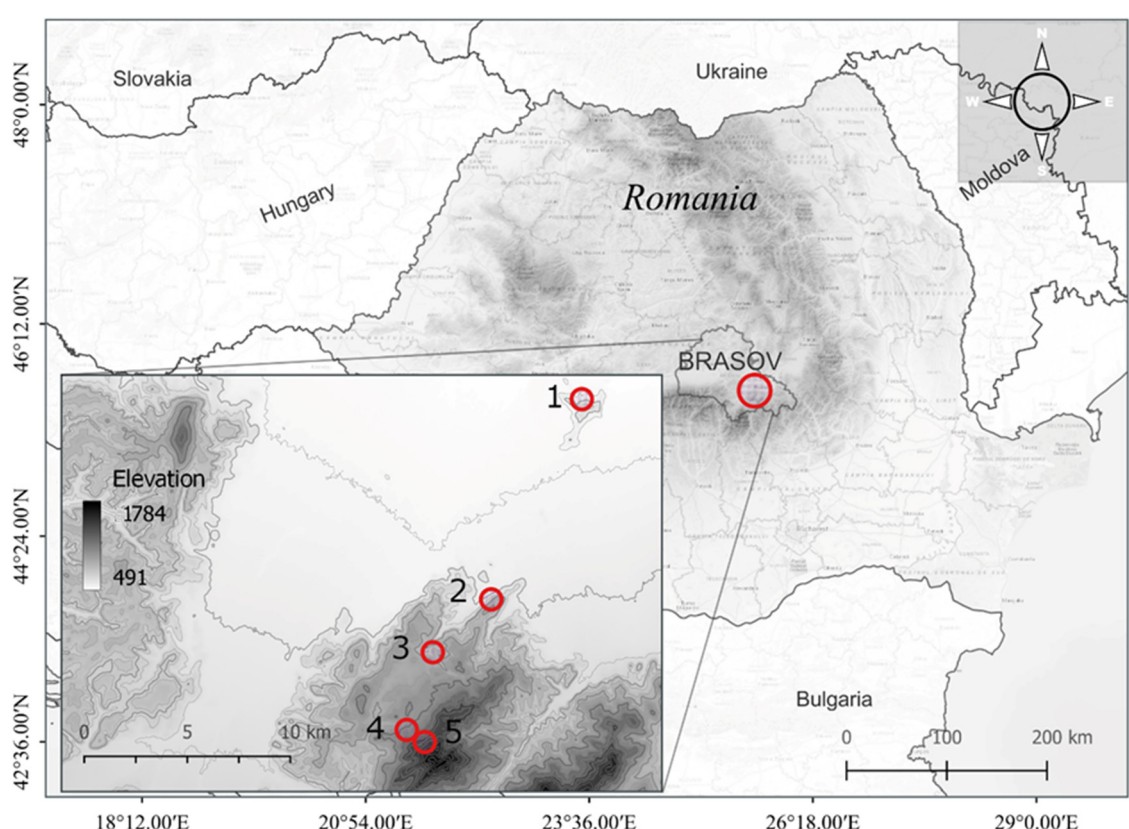

**Figure 1.** The geographic location of the studied European beech (*Fagus sylvatica* L.) populations across the altitudinal gradient from the south-eastern Carpathian Mountains (1—Lempes, 2—Tampa, 3—Solomon, 4—P. Lupului, and 5—Ruia).

**Table 1.** The geographic location of the studied European beech (*Fagus sylvatica* L.) populations from the south-eastern Carpathian Mountains and their altitude range.

| Population ID | Geographic Coordinates | Altitude Range (m) |
|---|---|---|
| Lempes | 45°43′34.88″ N<br>25°39′30.66″ E | 550–650 |
| Tampa | 45°38′18.86″ N<br>25°35′38.56″ E | 650–750 |
| Solomon | 45°36′59.75″ N<br>25°33′39.87″ E | 800–1000 |
| P. Lupului | 45°34′54.64″ N<br>25°32′36.43″ E | 1000–1150 |
| Ruia | 45°34′25.41″ N<br>25°33′11.67″ E | 1300–1450 |

### 2.2. Phenological Data

Phenological observations from the field were performed based on the methodology proposed by Vitasse et al. [34]. Every population from each study site was visited twice a week, from April to June (spring phenology) and September to November (senescence). These observations were always carried out by the same observer, with the naked eye or using binoculars, approximately 15 m away from the tree. The leaf unfolding (LU) process was divided into four stages of development (Table 2). Each tree received a stage depending on the majority proportion (>50%) of the buds from the upper third of the crown at that moment. Each study site population received an LU stage based on the average of the estimated stages for the 30 sampled individuals. Further, the qualitative scale of the bud-opening and leaf-unfolding process was converted into a quantitative one (according to the range of the percentage of green cover; Table 2) to have higher precision in delimiting the stages (a finer scale) and to adapt it to the same unit of measure (%) as that of senescence.

**Table 2.** Leaf-unfolding stages linked to the observed leaf development and green cover range.

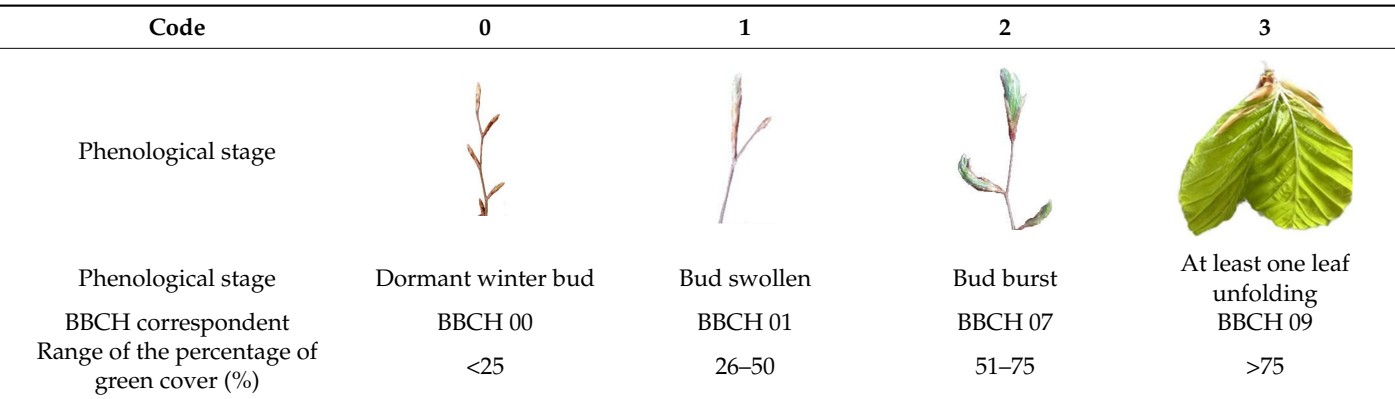

| Code | 0 | 1 | 2 | 3 |
|---|---|---|---|---|
| Phenological stage | Dormant winter bud | Bud swollen | Bud burst | At least one leaf unfolding |
| BBCH correspondent | BBCH 00 | BBCH 01 | BBCH 07 | BBCH 09 |
| Range of the percentage of green cover (%) | <25 | 26–50 | 51–75 | >75 |

The senescence (%CFL) was calculated based on two variables, the percentage of colored leaves and the percentage of missing (fallen) leaves, according to the following formula (1):

$$\%CFL = (\%CL \times (100 - \%FL))/100 + \%FL \tag{1}$$

where %CFL is the % of colored or fallen leaves (senescence), %CL is the % of colored leaves, and %FL is the % of fallen leaves.

The start of an individual's growing season was considered when it reached stage 2 (bud burst) according to the bud-opening and leaf-unfolding process [35,36]. The reporting in the growing season at the bud burst stage was carried out because this was the first visible one from the observer's level identified at the time of field observations [37]. This

stage also has the equivalent with the same name on the BBCH scale [38]. The start of the senescence was considered when the leaves started to yellow.

### 2.3. Meteorological Data

The process of acquiring raw meteorological data differed for each of the two monitoring years. In the first year, 2021, raw data were extracted from a database from meteorological stations near the plots where the studies were performed. In the second year, 2022, three temperature and relative humidity sensors were installed inside the stand, in representative points, for each study site (two HOBO loggers and one iButton logger). Each sensor was calibrated to record temperature and relative humidity values at a frequency of 30 min (48 values/day), later being compared with those from the nearby meteorological stations. Subsequently, these raw data were processed, and several meteorological indices were calculated based on them. The average daily temperature (°C) and relative humidity (%) values were calculated as the average of the 48 daily measurements recorded at a frequency of 30 min. From these 48 daily measurements (both for temperature and relative humidity), the maximum and minimum values were selected for calculating the maximum and minimum daily temperature ($T\_max$ and $T\_min$) and maximum and minimum daily relative humidity ($RH\_max$ and $RH\_min$). GDD (growing degree days) was used as a meteorological indicator for spring phenology, and it was calculated by subtracting the thresholds of 0 °C (GDD_0), 5 °C (GDD_5), and 10 °C (GDD_10) from the daily average temperature values. SDD (senescence degree days) was used as a meteorological indicator for autumn phenology, and it was calculated by subtracting the thresholds of 0 °C from the daily average temperature values.

### 2.4. Data Analysis

We performed a normality test and used variance analysis (ANOVA) to test for differences between populations in spring and autumn phenology. For the variance study, changes along the altitudinal gradient were considered according to each phenophase stage (bud burst, yellowing of the leaves) for the two years of monitoring. We also tested for correlations in phenological sensitivity to meteorological indicators (temperature, GDD, RH). To test for significant differences between the populations, we used a *t*-test.

All statistical analyses were computed in R software v. 4.3.1 [39]. The results were graphically displayed using the "ggplot2" and "corrplot" packages.

## 3. Results

### 3.1. Phenological Data

#### 3.1.1. Spring Phenology

Using the methodology of Vitasse et al. [34], we monitored the phenophases during the spring and autumn of 2021 and 2022. The duration amplitude in the bud burst stage was similar between the populations in 2021 and 2022 (29 days). In relative terms, individuals located at the highest altitude (Ruia) need 28 (23%) more days to start the growing season than those from the lowest altitude (Lempes). Significant differences exist in the dynamic of spring foliage phenology between these two years (Figure 2). In 2021, during the transition between the first phenophases, the stages' duration was faster than in 2022, at 2–3 days (7 days for the individuals at the maximum altitude). In the second year, the phenophases lasted longer, at 5–7 days (3 days for the individuals at the minimum altitude), representing a slower dynamic of the entire process starting in the growing season (10–13 days). There was also the same delay of 29 days between the two populations located at the extremities of the study area. Between the two monitored years, there was a difference of 14 days at the start of the growing season of the same populations.

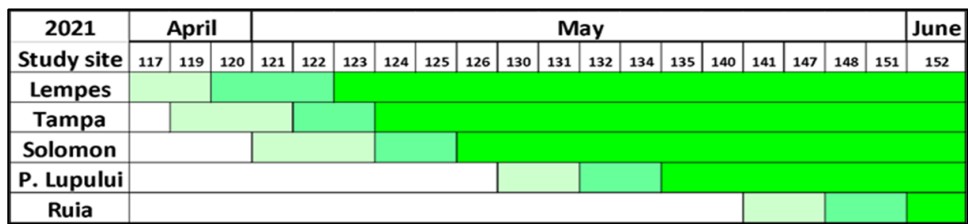

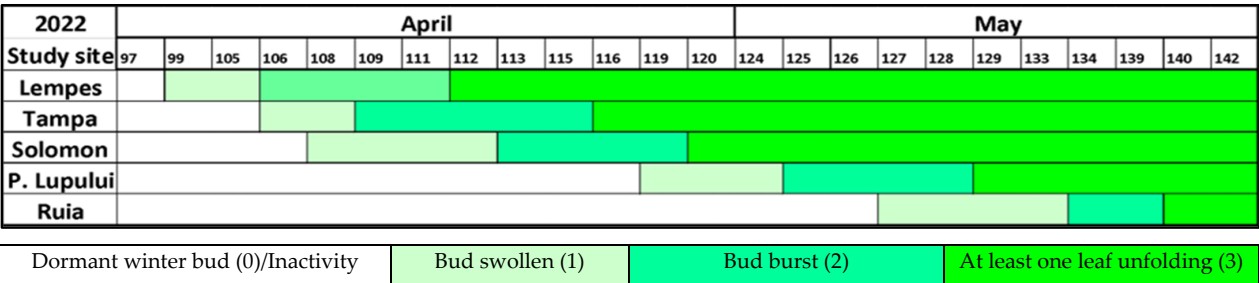

| | Dormant winter bud (0)/Inactivity | Bud swollen (1) | Bud burst (2) | At least one leaf unfolding (3) |
|---|---|---|---|---|

**Figure 2.** The dynamic of spring foliage phenology in 2021 and 2022 at each study site, based on mean values, along D.O.Y (day of the year).

Interpopulation variation (Figure 3) was significant in all cases ($p < 0.0001$ ANOVA test). The start of each phenophase at the population level varied according to the altitudinal gradient. There were significant differences between the two monitored years (ANOVA, $p < 0.0001$) in reaching the specific stage of each phenophase.

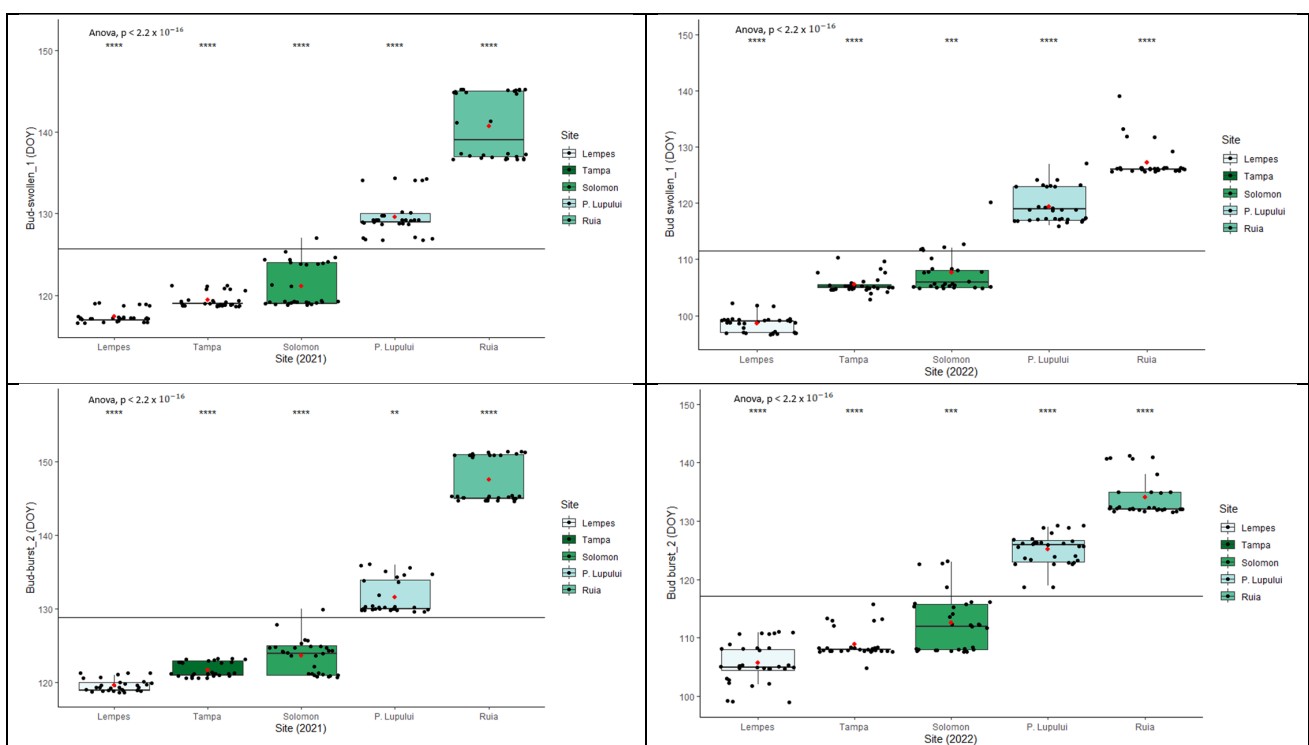

**Figure 3.** *Cont.*

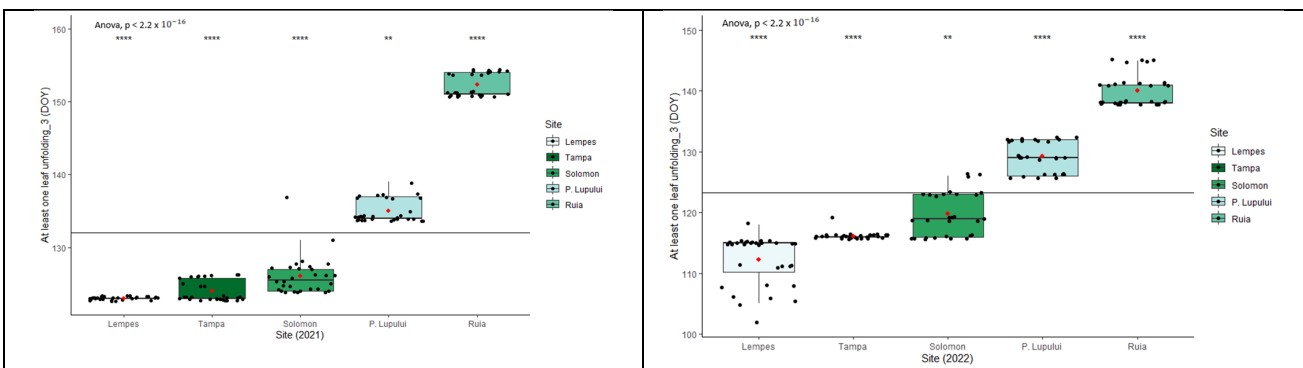

**Figure 3.** Intra- and interpopulation variation in European beech for each spring's phenophase in the two monitored years (****—$p \le 0.0001$, ***—$p \le 0.001$ and **—$p \le 0.01$).

### 3.1.2. Autumn Phenology/Senescence

The dynamic of autumn foliage phenology varied during the two monitored years (Figure 4). In this case, the senescence was estimated by quantifying the two leading indicators of phenophases, yellowing and falling of the leaves. The phenophase of leaf yellowing took a more extended period in 2021 (21–32 days) than in 2022 (18–25 days), with a difference between them of 16%–28%, proportional to the increase in altitude. As in the case of spring phenology, the variation was directly proportional to the elevational gradient, with some exceptions (2021) found in the sites with special stationary conditions (wind exposure, stand density).

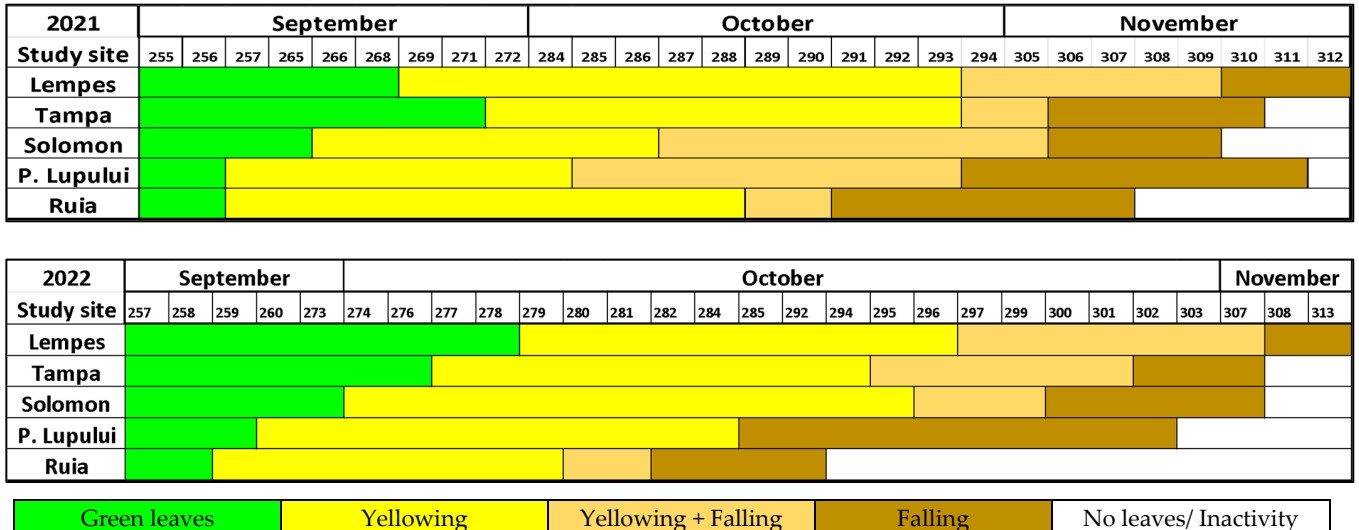

**Figure 4.** The dynamic of autumn foliage phenology during 2021 and 2022 at each study site, based on mean values, along D.O.Y (day of the year).

Interpopulation variation (Figure 5) was significant in all cases ($p < 0.0001$). In the same stationary conditions (same site), individuals with an early onset in the growing season showed the same early behavior in senescence. There were significant differences between the two monitored years (ANOVA, $p < 0.0001$) in reaching the specific stage of each phenophase.

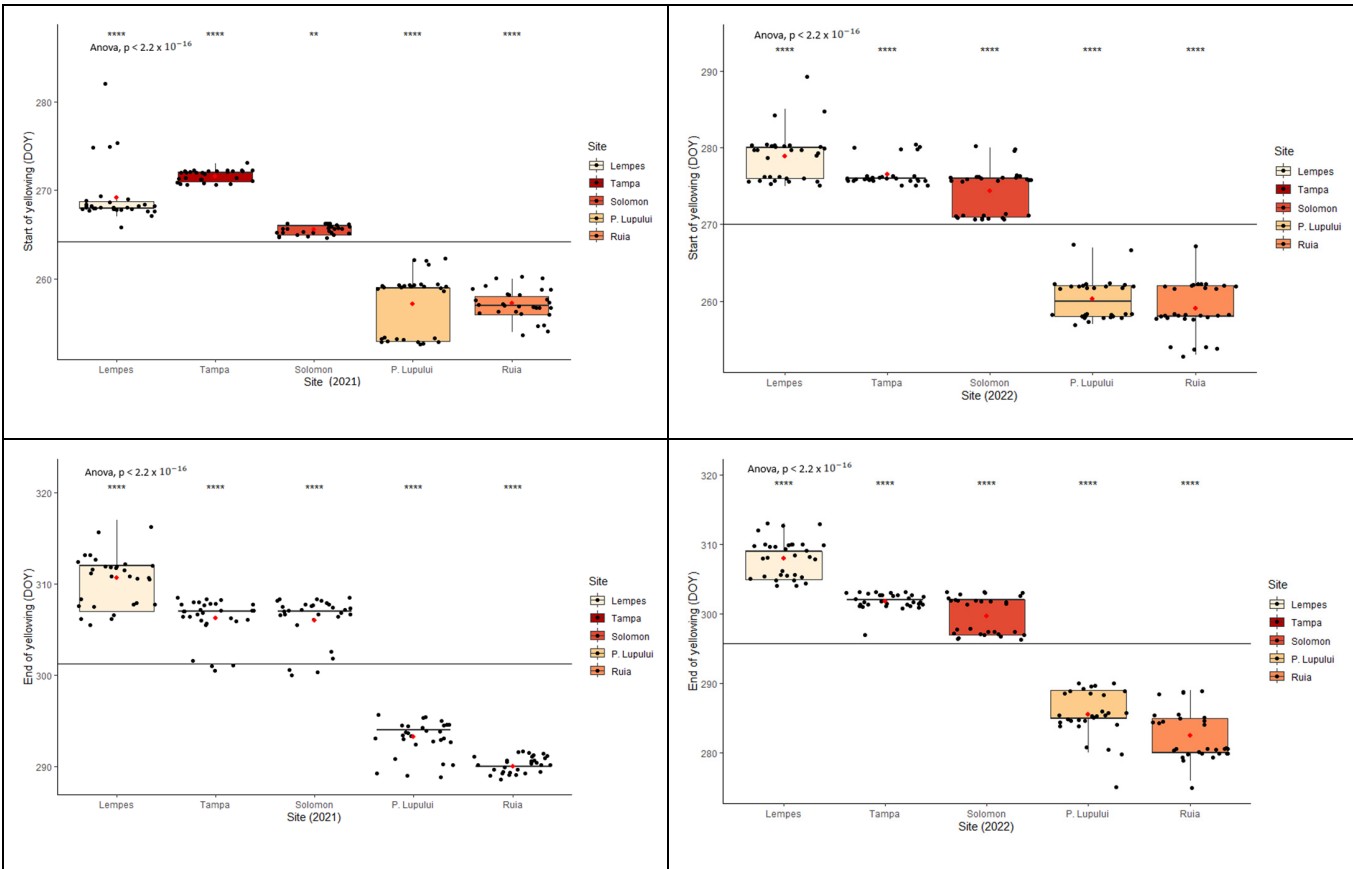

**Figure 5.** Intra- and interpopulation variation in European beech for the yellowing of leaves phenophase in the two monitored years (****—$p \le 0.0001$ and **—$p \le 0.01$).

The analysis of leaf fall of the individuals from these five study sites revealed significant interpopulation variation (Figure 6), with significant differences between all populations ($p < 0.0001$ ANOVA test). There were significant differences between the two monitored years (ANOVA, $p < 0.0001$) in reaching the specific stage of each phenophase.

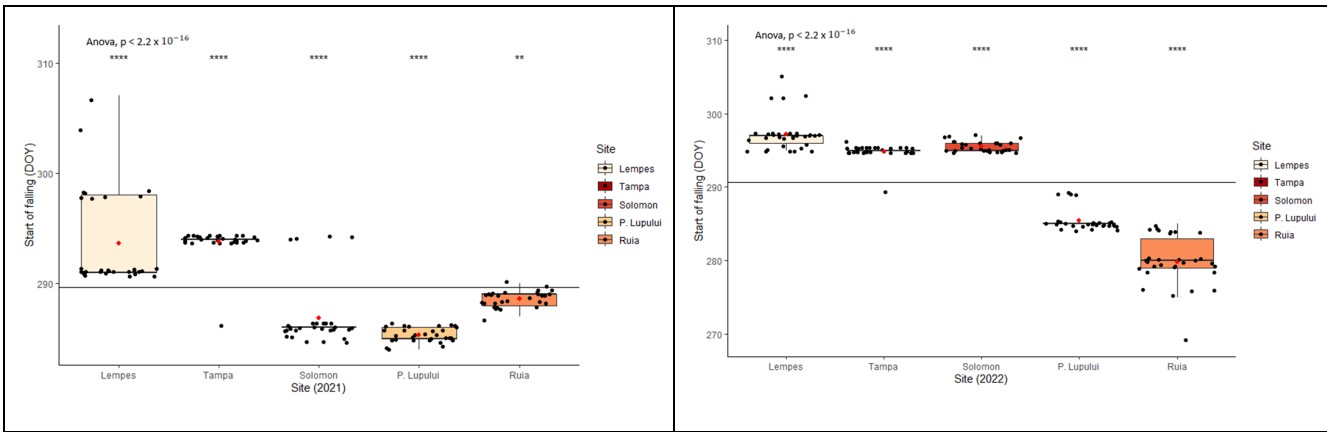

**Figure 6.** *Cont*.

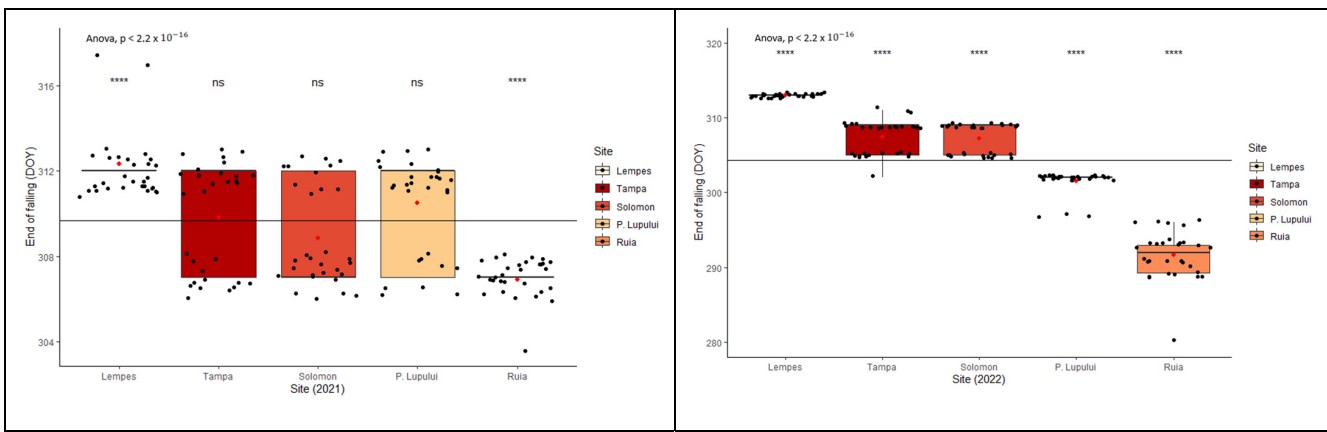

**Figure 6.** Intra- and interpopulation variation in European beech for the falling of leaves phenophase in the two monitored years (****—$p \leq 0.0001$, **—$p \leq 0.01$ and ns—$p > 0.05$).

### 3.1.3. Length of Growing Season

The length of the growing season of individuals (defined as a period between the average onset of bud burst and leaf-yellowing phenophases) varied during the two years of monitoring (Figure 7). There were significant differences between the two monitored years (ANOVA, $p < 0.0001$). An increase in altitude causes a later bud burst, a more premature onset of senescence, and a shorter growing season. A delay in the start of the spring phenophases, such as in the case of 2021, implies a shorter growing season. The average growing season duration ranged from 109 to 150 days in 2021 and 125 to 176 days in 2022. On average, about a 37% shorter growing season was observed at the highest altitudes compared to the lowest in both monitored years, and about a 14% longer growing season in 2022 compared with 2021.

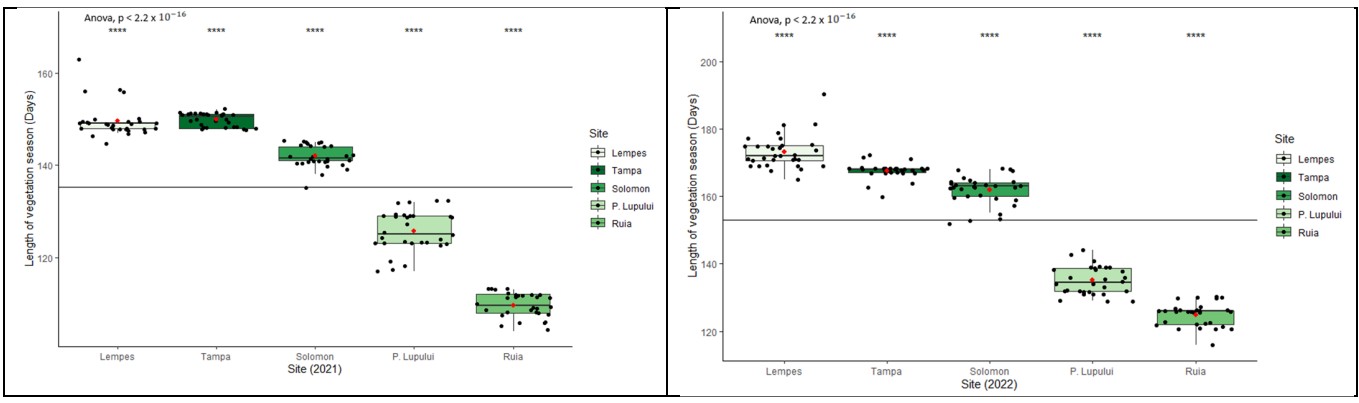

**Figure 7.** Intra- and interpopulation variation in European beech for the length of growing season in the two monitored years (****—$p \leq 0.0001$).

### 3.2. Meteorological Data

Historical temperature records were obtained from 1970 to 2000 from the WorldClim database [40] (Figure 8) for each study site. Compared with the values of 2021 and 2022, there is a trend of increasing annual temperatures. Along this elevational gradient, the temperature decreases by 0.5 °C with 100 m increasing altitude A linear regression model was fitted to the temperature variation along this altitudinal gradient (y = −0.99x + 11.03; $R^2 = 0.9486$).

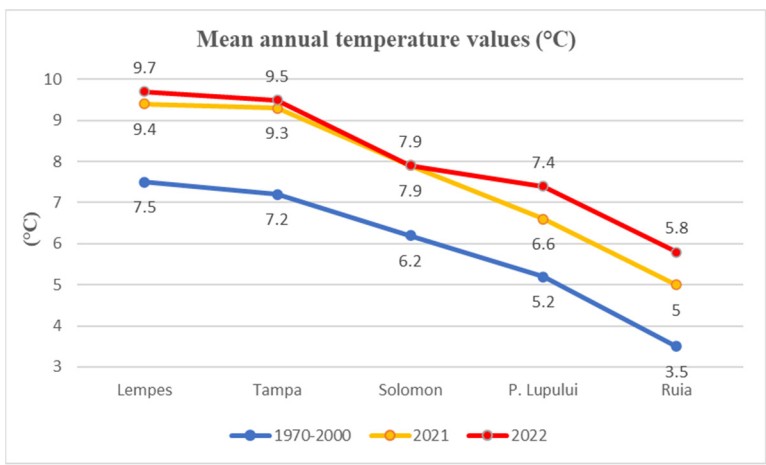

**Figure 8.** Mean temperature values for each study site for 1970–2000 compared with monitored years of 2021 and 2022.

*3.3. Relationships between Phenological and Meteorological Data*

The association between phenological and meteorological data shows that the daily average temperature triggers the start of the growing season the most (Figure 9). The other indicators (daily maximum and minimum values, growing degree days with the threshold of 0 °C, 5 °C, and 10 °C) obtained equally good values (>0.8), the explanation being their calculation method, which is also based on the daily average temperature. The correlation of phenological data with those of humidity (daily average value and daily maximum value) obtained a very low Pearson correlation value of −0.22 and −0.10, respectively.

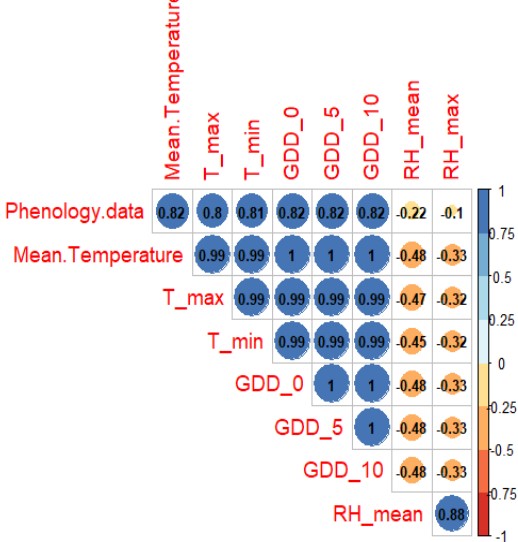

**Figure 9.** Correlogram of the Pearson correlation between phenological and meteorological data. The significance of the probability hypothesis (correlation) is indicated by the Pearson correlation value, the color (according to the color gradient from the right), and the size of the circles.

Temperature is the triggering factor for bud burst. When the thermal threshold of 10 °C is exceeded, bud burst occurs, but this condition is complementary to exceeding a specific accumulation of GDD (with a threshold of 0 °C). During the two monitored years, flush occurred after accumulating at least 60 GDD in the last 7 days (Figure 10).

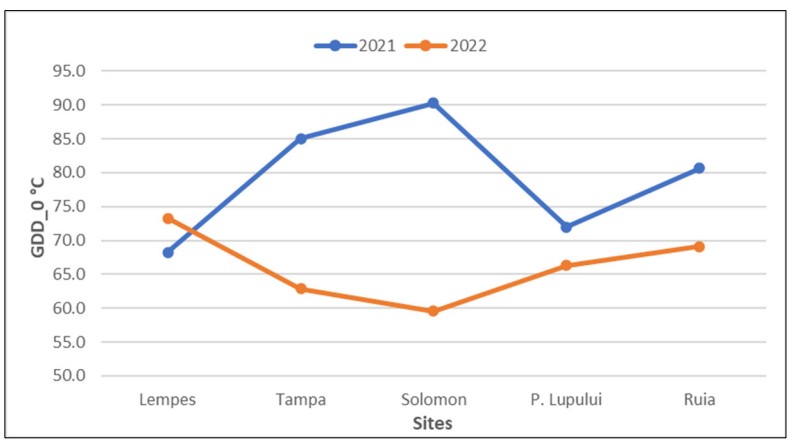

**Figure 10.** Cumulative GDD (growing degree days) with a threshold of 0 °C in the two monitored years for bud burst (associated with the start of the growing season) in the last seven days for each population (ascending altitudinal order).

Senescence is also influenced by temperature. The first phenophase of autumn phenology, the yellowing of the leaves, is triggered by an accumulation of at least 72 SDD with a threshold of 0 °C in the last 7 days (Table 3).

**Table 3.** Cumulative SDD (senescence degree days) with a threshold of 0 °C in the two monitored years on European beech reaching the yellowing of the leaves phenophase (associated with the start of the senescence) in the last seven days for each population (ascending altitudinal order).

| Study Site | 2021 | 2022 |
|---|---|---|
| Lempes | 72.14 | 81.49 |
| Tampa | 80.25 | 89.79 |
| Solomon | 72.87 | 85.47 |
| P. Lupului | 86.55 | 75.86 |
| Ruia | 78.97 | 78.35 |

The second phenophase of senescence, leaf fall, is not influenced to the same extent as the yellowing of the leaves by the accumulation of the average temperatures of the last seven days nor by the appearance of frost (the decrease in the thermal threshold of 0 °C). It is still inversely proportional to the altitudinal gradient (Figure 11).

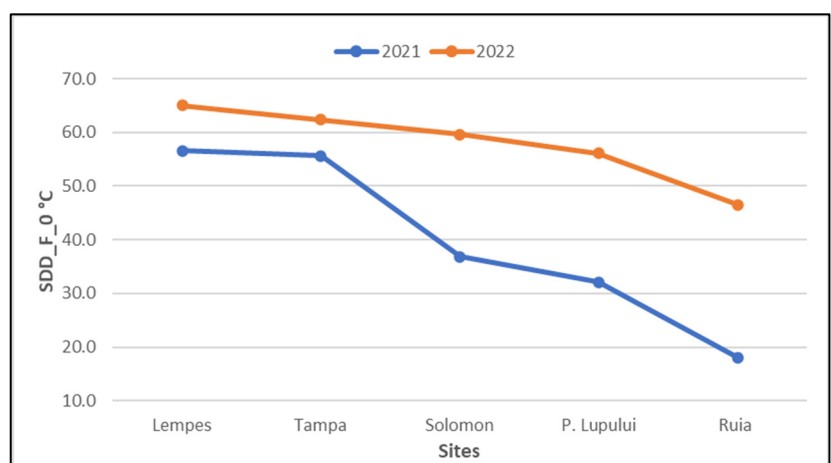

**Figure 11.** Cumulative SDD (senescence degree days) with a threshold of 0 °C in the two monitored years on European beech reaching falling of the leaves phenophase in the last seven days for each population (ascending altitudinal order).

## 4. Discussion

This study aimed to bring forth new information about European beech phenological behavior along an elevational gradient and predict the effects of climate change on this species' future distributions in the south-eastern Carpathians [30]. We performed phenological observations in the field and correlated the phenological data with the meteorological data to see which climatic indicator (temperature or humidity) predominantly influences the phenology of this species.

The two monitored years were phenologically different: in 2021, the start of the growing season occurred with a delay, associated with a faster transition between phenophases; this also implies a shorter growing season, compared to 2022, where the bud burst occurred earlier, the transition from one phenophase to another was slower, and the growing season was more extended [41].

The onset of spring phenology occurs when the daily temperature exceeds the thermal threshold of 10 °C [33], confirming the fact that reaching the leaf-unfolding phenophase in European beech is strongly influenced by temperature [9,29,42] and less sensitive to micro-topographical factors [43]. Our study shows that a day or two in which the daily average temperature exceeds the threshold of 10 °C is not enough to trigger this process. A specific accumulation of at least 60 GDD (growing degree days) in the last 7 days is necessary. Altitude is the main macroecological factor influencing leaf unfolding [44] because it also involves temperature variation.

Our results suggest that temperature is also the main driver of senescence, more precisely in its first phenophase, the yellowing of the leaves, confirming other similar studies [45,46]. However, there is still no information about autumn phenophases and their correlation with environmental factors [47,48]. It may be more difficult to detect the correlation between senescence and other meteorological factors, mainly due to the less precise quantification of the yellowing of the leaves from the upper third of the crown [49] and the less concrete delimitation of the influence on the yellowing of the leaves caused by senescence or drought. Our results showed that marcescence is not influenced to the same extent as the yellowing of the leaves by the accumulation of a certain SDD in the last seven days, nor by the appearance of frost (a decrease in the thermal threshold of 0 °C), and that it can be strongly related to micro-topographical factors (the location of the tree/stand and its exposure to air currents/wind).

The variation between individuals of the same populations is wider in the case of senescence, compared with spring phenology (see also Vilhar et al. [50]), due to the existence of several other factors that influence this process and the lack of precision in delimiting the impact of each one.

The spatial variability of temperature is related to the elevational gradient, with increasing altitude, the temperature decreases [36].

## 5. Conclusions

Monitoring tree phenology and analyzing these data are essential for predicting the effects of climate change on forest ecosystems. The association of phenology with meteorological data confirmed that temperature is the triggering factor for both spring phenology and senescence. Our study showed a variation in phenological stages on European beech along an altitudinal gradient, with individuals at low altitudes exhibiting an earlier onset of bud burst and a faster transition through the phenophases of spring phenology. The variation in the same individuals through senescence is also due to altitudinal and thermal differences.

The results of this study have important implications for understanding the phenological responses of European beech to climate change. As temperatures continue to increase, the start of the growing season is expected to occur earlier, while senescence is expected to be delayed. This could lead to a longer growing season overall, but it could also increase the risk of drought stress during the early stages of the growing season.

However, further investigation into phenological patterns is needed to develop models to predict how these factors will have implications. This may contribute to creating a strategy for beech forest management practices and conservation.

**Author Contributions:** M.I.C.C. and A.L.C. designed the experiment. M.I.C.C. performed the fieldwork. M.I.C.C., E.C., G.R.R., V.D.P. and E.B. analyzed the data. M.I.C.C. wrote the manuscript. E.C., D.C., G.R.R., V.D.P., E.B., O.G.Z., O.G. and A.L.C. reviewed and edited the manuscript. All authors contributed to the article and approved the submitted version. All authors have read and agreed to the published version of the manuscript.

**Funding:** This research was conducted within the projects PN 23090102 and 34 PFE/30.12.2021 "Increasing the institutional capacity and performance of INCDS 'Marin Drăcea' in the activity of RDI—CresPerfInst" funded by the Ministry of Research, Innovation and Digitalization of Romania. Additional funding was provided by the Federal Ministry of Food and Agriculture of Germany within the project "DroughtMarkers" (FNR, Waldklimafonds, Reference number: 2218WK43B4).

**Data Availability Statement:** The data used in this work are available from the corresponding author upon reasonable request.

**Acknowledgments:** We want to acknowledge "Transilvania" University from Brasov, "Marin Drăcea" National Institute for Research and Development in Forestry (INCDS), the administration of the National Agency for Natural Protected Areas (ANANP), and RNP Romsilva. We also wish to thank our colleagues Florentina Chira and Costel-Ștefan Mantale for their help with the fieldwork.

**Conflicts of Interest:** The authors declare no conflicts of interest.

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
