# Peer review of "Large Differences in Bud Burst and Senescence between Low- and High-Altitude European Beech Populations along an Altitudinal Transect in the South-Eastern Carpathians"

_forests, doi:10.3390/f15030468_

Round 1

Reviewer 1 Report

Comments and Suggestions for Authors

The manuscript tried to explore the regulations between altitude, latitude and folia phenology, and have found some interesting results. It tells a good story while the INTRODUCTION part needs reconstruction. For example,

L49-51, Not clear

L66-69, It will be better if this information could be placed in the 2nd place of the paragraph.

L89-91, Move to L72

L187-189, Delete it.

Climate warming has advanced the spring phenology and delayed the autumn phenology of the European forest. Temperature was considered as the key driver of them. The present manuscript explored the regulations between altitude and folia phenology of European beech (bud burst and leaf yellowing), and have confirmed again that temperature driven the variation of the spring and autumn phenology of the special species along the altitude. These species level results are helpful for forest managers to develop forest management strategies.

The manuscript tells a good story while there are some issues still need to be addressed.

L159-160: How near the meteorological stations are from each of the plots? Are those data representative compared to the data from the inside ones. Add some explanations or a meteorological station distribution graph. Is there any missing data? If so, how is it supplemented?

L211-212: There are significant differences between the two monitored years in bud swollen and bud burst. What’s the reason? Are there some correlations to the temperature (GDD) difference between the two years just like you have showed in Figure 10? Compared to Figure 8(mean annual temperature), there were contrasting differences in GDD between the two years in Figure 10.

L281-283: Figure 9, have you tried the non-linear correlations between phenological events and the meteorological data since usually they are non-linear relationships.

Senescence is also influenced by temperature. Just like the spring phenology, I miss some inter-annual analysis about the cause of it. There are such big differences between the two years in SDD_F_0 °C in Figure 11.

A suggestion, according the present results, temperature is the triggering factor of the folia phenology, then, the causes of the interannual variation of phenology should be consistent with the causes of the variation of phenology with altitude gradient. Could the authors test this?

Best,

Author Response

We thank reviewer 1 for his critical comments that helped to improve our paper. Specific comments to requests follow below.

Reviewer 2 Report

Comments and Suggestions for Authors

The manuscript is appropriate to be published in Forests. It shows results of 2-year study on the dependence of leaf phenology in Fagus sylvatica on thermal conditions. The data are satisfying and the manuscript is generally well written. However, some parts should be improved, including methodological issues and graphs. All my remarks are listed below.

  1. Keywords: “bud burst” and “senescence” are not necessary; keywords are a tool which allow for widdening the range of words connected with the subject of the described research in the case when all necessary words are not included in the title.

  2. M&M, lines 122-123: Latin names of tree species instead of English ones would be preferable, particularly, taking into account non-European readers.

  3. M&M, lines 125-126, Figure 1 caption: please specify the names of the locations in the Figure caption.

  4. M&M, chapter 2.2: why don't the Authors use names of phenological stages according to the BBCH scale? This is a well-known scale used in numerous publications. Please include these names in the manuscript.

  5. M&M, chapter 2.4, Data analysis: did the Authors performed a normality test for the data? If normality is missing non-parametric test, e.g. Kruskal-Wallis should be performed instead of ANOVA, and post-hoc Dunn's test for comparisons between populations.

  6. Results, lines 187-189: these lines should be removed.

  7. Results, Figures 3, 5, 6, 7: please use larger fonts and characters in the graphs, at least twice (or more). Names and numbers in the graphs are illegible.

Author Response

We thank reviewer 2 for his critical comments that helped to improve our paper. Specific comments to requests follow below.
